# Should Perirectal Swab Culture Be Performed in Cases Admitted to the Neonatal Intensive Care Unit? Lessons Learned from the Neonatal Intensive Care Unit

**DOI:** 10.3390/children10020187

**Published:** 2023-01-19

**Authors:** Aysen Orman, Yalcin Celik, Guliz Evik, Gulden Ersöz, Necdet Kuyucu, Berfin Ozgokce Ozmen

**Affiliations:** 1Division of Neonatology, Department of Pediatrics, Mersin University School of Medicine, Mersin 33110, Türkiye; 2Department of Clinical Microbiology and Infectious Diseases, Mersin University School of Medicine, Mersin 33110, Türkiye; 3Department of Pediatric Infectious Disease, Mersin University School of Medicine, Mersin 33110, Türkiye

**Keywords:** perirectal swab, carbapenem, resistance, enterobacterales, vancomycin, enterococci, neonatal unit

## Abstract

Serial perirectal swabs are used to identify colonization of multidrug-resistant bacteria and prevent spread. The purpose of this study was to determine colonization with carbapenem-resistant Enterobacterales (CRE) and vancomycin-resistant Enterococci (VRE). An additional purpose was to establish whether sepsis and epidemic associated with these factors were present in the neonatal intensive care unit (NICU), to which infants with hospital stays exceeding 48 h in an external healthcare center NICU were admitted. Perirectal swab samples were collected in the first 24 h by a trained infection nurse using sterile cotton swabs moistened with 0.9% NaCl from patients admitted to our unit after hospitalization exceeding 48 h in an external center. The primary outcome was positivity in perirectal swab cultures, and the secondary outcomes were whether this caused invasive infection and significant NICU outbreaks. A total of 125 newborns meeting the study criteria referred from external healthcare centers between January 2018 and January 2022 were enrolled. Analysis revealed that CRE constituted 27.2% of perirectal swab positivity and VRE 4.8%, and that one in every 4.4 infants included in the study exhibited perirectal swab positivity. The detection of colonization by these microorganisms, and including them within the scope of surveillance, is an important factor in the prevention of NICU epidemics.

## 1. Introduction

The World Health Organization (WHO) has described antimicrobial resistance as one of the most severe threats to public health in the last decade, with impacts expected to worsen in the future [1]. Multidrug-resistant (MDR) microorganisms are the main factors in epidemics in neonatal intensive care units (NICUs) [2] Sepsis due to carbapenem-resistant *Enterobacterales* (CRE) and vancomycin-resistant *Enterococci* (VRE) is a matter of particular concern [3,4]. Based on population-level studies over the past two decades, the global estimated incidence of mortality from neonatal sepsis ranges from 11% to 19% [5]. Due to their increasing incidence, MDR microorganisms are estimated to be responsible for a significant proportion of deaths from neonatal sepsis [6]. It is therefore important to maintain a high level of concern, take appropriate precautions, and identify outbreaks promptly [7,8]. To prevent infection or colonization by MDR organisms, it is necessary to combine hand hygiene monitoring systems, MDR microorganism surveillance, contact isolation measures (e.g., single room isolation or group cohorts of cases exposed to bacterial colonization), environmental cleanliness, prudent antimicrobial agent use, and antibiotic management strategies [9,10,11]. The serial screening of selected body sites with swab cultures is a common practice in NICUs for monitoring horizontal pathogen spread and preventing the occurrence of septic outbreaks [7]. Over time, changes have been observed in multi-drug resistant infectious agents worldwide and in Turkey, and new antibiotics effective against Gram-positive have entered into use. Due to these new changes, there have been doubts about whether surveillance prevents increases in VRE and causes changes in mortality and morbidity rates. Although the increase in the incidence of carbapenem-resistant *Enterobacterales*, *Pseudomonas aeruginosa*, and *Acinetobacter* species among MDR microorganism became a serious problem, the number of Gram-positive microorganisms decreased over time [12]. Despite this decrease in Gram-positive agents, VRE still causes widespread colonization in the early period and invasive and difficult-to-treat infections in critically ill patients by causing in-hospital transmission from patients and the environment [13]. Therefore, rectal swab collection may be recommended as part of surveillance in critically hospitalized patients [14].

Considering the risk of progression to active disease in non-symptomatic carriers among newborns, an effective methodology is therefore needed for the management of outbreaks, particularly in NICUs. There is no national guideline for the routine screening of newborns for MDR organisms in NICUs in Turkey. Each hospital therefore produces its own infection prevention policy. In addition, collecting perirectal swabs from all infants treated in NICUs increases costs and workloads, and also causes unnecessary antibiotic use. However, whether the screening strategy is a cost-effective measure remains a matter of debate [15]. Institutions in developing countries should identify their own high-risk patients, and screening priorities should be based on infection prevalence and hospital finances [16].

In this study, VRE and CRE colonization were routinely investigated in patients admitted from an external center and hospitalized in our neonatal intensive care unit. The purpose of this study was to prevent the spread of VRE and CRE by applying contact isolation to screened patients. This is the first pilot study to evaluate the spread and epidemic of sepsis within the NICU using this method. 

## 2. Materials and Methods

### 2.1. Study Design, Setting, and Population

The study was conducted in accordance with the Declaration of Helsinki and approved by the Mersin University Clinical Research Ethics Committee (Decision No: 2022/437, Date: 25 May 2022). The study was conducted in the Mersin University Faculty of Medicine NICU between 1 January 2018, and 1 January 2022. Our hospital’s NICU provides quaternary care and has a total bed capacity of 27, with a nurse-patient ratio of 1/3. There are 25 beds in an open area in that unit, plus two single isolation rooms. Two neonatology specialists (a professor of medicine and an assistant professor of medicine), a neonatology intern and three pediatric interns work in our NICU. The doctor patient ratio is 1/7.

In line with our clinic’s standard procedures, perirectal swabs and blood culture samples were collected for VRE and CRE from infants hospitalized in another healthcare center for longer than 48 h. Contact isolation was applied by regarding these as positive until the results were obtained. When the perirectal swab was positive, patients were taken to single rooms for contact isolation or to the isolation room of patients infected with the same microorganism. Nurses and allied health personnel responsible for the care of these patients were also included in the cohort. These were cohorted with the nurses and assistant staff responsible for patient care. In the event of VRE/CRE growth in one blood culture or two clinical isolates within 10 days because of the presence of a patient colonized/infected with VRE/CRE, surveillance screening was initiated to include all hospitalized patients with weekly screening cultures taken for at least two consecutive weeks of new positive growth. These continued until no positivity was identified. Scanning resumed if positivity was again detected in clinical samples. All patients hospitalized on a ward included in the surveillance were screened once a week by the infection control team on a predetermined day. Perirectal cultures were cultivated in selective media at the bedside and studied in the laboratory. Patients with positive perirectal swabs were followed-up until discharge in order to monitor the development of infection. Since VRE can survive on inanimate surfaces for very extended periods (such as commodes, dining tables, door handles, doctor and nurse observation tables, monitors, and ventilators), the samples were collected when VRE growth was detected in perirectal culture. The surfaces from which cultures were to be taken was determined by the infection control team. The procedure was performed by the same team. Summarized in study design, setting, and population Figure 1.

CRE and/or VRE colonization in this study was defined as cases with positive perirectal swab samples without signs or symptoms of infection in the newborn. CRE and/or VRE infection was defined as a positive sample from a normally sterile site (blood and cerebrospinal fluid) with signs and symptoms of infection. The definition of outbreak was based on isolates of the same species from two or more sterile sites with the same antibiogram from different patients over two weeks.

#### 2.1.1. Case Definition and Inclusion Criteria

Patients with hospital stays longer than 48 h in an external healthcare center admitted to our NICU following referral for advanced intensive care, and with perirectal swab cultures collected for CRE and VRE within the first 24 h after admission to the unit were included in the study. Infants with hospital stays shorter than 48 h in an external healthcare center and/or with congenital anomalies that prevented perirectal swab collection (such as anal atresia) were excluded from the study. Cases with missing or insufficient data were also not included in this retrospective study.

#### 2.1.2. Data Collection

Demographic (gestational age, birth weight, sex, age in days at referral from an external healthcare center) and anamnestic/clinical data (duration of antibiotic treatment and which antibiotic treatment was given in the other healthcare center, initial hospitalization diagnoses, and presence of sepsis at first admission, hospitalization weight after admission to the unit, mechanical ventilator monitoring presence and duration, antibiotic treatment and duration, blood culture results sent simultaneously with the perirectal swab, causative microorganisms (if any), perirectal swab culture results for CRE and VRE, and morbidity and mortality data) were analyzed retrospectively and recorded.

#### 2.1.3. Outcomes

The primary outcomes were positivity in swab cultures, and the secondary outcomes were whether this caused invasive infection and epidemic in the unit.

### 2.2. Microbiological Evaluation

#### 2.2.1. Isolation of Bacterial Isolates

Perirectal swab samples for CRE and VRE were collected from patients who met the inclusion criteria within the first 24 h of admission to our unit by a trained infection nurse using a sterile cotton swab moistened with 0.9% NaCl. Two hundred fifty swab cultures were thus obtained from 125 patients. The swab was gently rotated in the perirectal area and placed into selective liquid medium. The collected samples were immediately transferred to the Infectious Diseases Laboratory. Daily turbidity control for isolation of the factor VRE at the bedside is performed using Mueller-Hinton Broth (MHB) containing 6 mg/mL vancomycin and 64 mg/mL ceftazidime as a selective liquid medium, followed by incubation at 37 °C for 24–72 h in the laboratory. When turbidity occurs, passages are made from the tubes to D-Coccosel agar that contains 6 mg/mL vancomycin and 64 mg ceftazidime as a selective medium for VRE. After the suspicious colonies have been kept for up to 72 h, the process is completed by performing the vancomycin E-test with Mueller-Hinton Agar to verify the outcomes.

The perirectal swab samples were placed in 5–10 mL MHB containing a 10 µg ertapenem disc and incubated for 72 h for CRE. Daily turbidity checks were performed. When turbidity occurred, a passage was made on the EMB agar with a selective meropenem or ertapenem disc for Gram-negative bacteria producing carbapenemase. All these processes were analyzed retrospectively at archive scanning, and non-fermentative growths and growths outside the antibiotic zone were excluded from the study.

#### 2.2.2. Statistical Analysis

The descriptive variables were first examined in both independent groups. The normality of distribution of continuous measurements was checked using the Kolmogorov-Smirnov test. The Mann-Whitney U-test was applied in comparisons between the two groups since the continuous variables were not normally distributed. The χ^2^ test was used to compare categorical variables. The results were analyzed on Statistical Package for Social Sciences version 17 software (SPSS, Chicago, IL, USA), and *p* values < 0.05 were considered significant.

## 3. Results

A total of 994 patients were admitted between January 2018 and January 2022 following referral from an external healthcare center. Only 125 patients were hospitalized in another center for longer than 48 h, and 125 newborns who met the inclusion criteria were enrolled in the study. During the study period (January 2018–January 2022), 28.8% growth was detected in 250 perirectal swab samples collected from infants who were hospitalized in another healthcare center for more than 48 h and then referred to us. Perirectal swab positivity was 27.2% (n = 34) CRE and 4.8% (n = 6) VRE. Perirectal swab positivity was thus detected in approximately one in every 4.4 infants. Perirectal swab results were divided into groups CRE positive and negative, and VRE positive and negative. The relationship between these groups and demographic data (sex, mode of delivery, weight at birth and hospitalization, length of hospitalization and antibiotic therapy in an external healthcare center, and referral from a public or private institution) was investigated. No statistically significant difference was found. *p* values are given in Table 1.

The effects of CRE and VRE growth results in perirectal swab on the clinical course after admission to the unit were also evaluated (Table 2). As shown in Table 2, the presence or absence of CRE in the perirectal swab had no effects on the length of hospital stay, but a statistically significant difference was detected with VRE scanning (*p* = 0.039). Considering this difference, VRE-negative patients at the perirectal swab (%50,4) were hospitalized for 20 days longer than those who were identified as positive. The diagnoses of VRE-negative patients hospitalized for more than 20 days involved respiratory causes (n = 10, 16.7%), metabolic causes (n = 10, 16.7%), surgical reasons (n = 21, 35%), asphyxia (n = 8, 13.3%), neonatal infection (n = 15, 7.6%), and hematological causes (n = 2, 3.3%). Six patients who were VRE-positive were not hospitalized with surgical, asphyxia, or metabolic disease diagnoses, but due to respiratory (n = 3, 50%), neonatal infection (n = 2, 33.3%), and hematological (n = 1, 16.7%) causes. Surgical causes, asphyxia, and metabolic diseases therefore appeared to be effective in prolonging hospitalization.

An examination of the factors reproducing in the blood culture of patients with definite diagnoses of sepsis with CRE-positivity in perirectal swab, revealed methicillin-resistant coagulase-negative *Staphylococci* (MRCNS) in two patients and carbapenem-sensitive *Klebsiella pneumoniae* in one. There was only one patient diagnosed with definite sepsis with VRE positivity in the perirectal swab, and carbapenem-sensitive *K. pneumoniae* grew in his blood culture. A statistically significant relationship was detected between positive VRE in perirectal swabs and procalcitonin (PCT) (*p* = 0.039). However, this was because of the differences in the VRE negative and positive numerical values. A statistically significant relationship was also detected between blood culture results and VRE positivity in perirectal swabs (*p* = 0.038). The patients with VRE positivity in perirectal swabs exhibited 16.7% *K. pneumoniae* growth in blood culture, while the VRE-negative patients had MRCNS (4.2%). When CRE and VRE in the unit were examined in terms of outbreak, no match was detected between the agents grown in blood culture and those grown in perirectal swab. No outbreak was detected during the study period. Hospitalization diagnoses according to perirectal swab culture results are summarized in Table 3. No statistically significant relationships were detected between diagnosis of disease and perirectal swab results in newborn infants.

## 4. Discussion

The emergence of multidrug-resistant organisms in neonatal sepsis has led to the need for new strategies, especially in neonates, who are highly vulnerable to outbreaks of healthcare-associated infections [17]. Perirectal active surveillance of MDR microorganism carriage can be considered an effective tool for early detection of unusual microbial pathogen patterns. Emerging evidence indicates an increased prevalence of MDR gram-negative (MDR-GN) colonization and neonatal MDR-GN infection in neonatal settings [6].

The prevalence of CRE was 27.2% in patients admitted to our NICU through referral from an external healthcare center. Prevalences of CRE of 21% in neonatal units in Vietnam, 5.8% in Portugal, 2.4% in Ethiopia, 1.8% in Morocco, and 1.6% in Algeria have been previously reported [18,19,20,21,22]. Unfortunately, the prevalence of CRE in this study appears to be higher than those in middle-income and developed countries [22,23]. This rate is worrying, and Turkey has been identified among the endemic countries [24]. Based on the results of this study, the use of broad-spectrum antibiotics could not be associated with CRE positivity. However, we think that the infrastructure and current problems of the units to which the patients were referred should be evaluated. For all these reasons, an urgent review of infection prevention and control measures appears to be required. 

Risk factors for CRE infection and/or colonization have been widely reported in adults, but more rarely in newborns. The length of hospital stay, orogastric nutrition, exposure to antipseudomonal antibiotics, previous surgical operation, and mechanical ventilation have been identified as separate risk factors for CRE colonization [25,26,27]. There are also studies reporting that maternal colonization may be important in neonatal gastrointestinal colonization [21,28,29]. In this study, descriptive features such as the exposure and duration of broad-spectrum antibiotics, the length of hospital stay, and mechanical ventilator follow-up, had no effect on colonization or non-colonization.

VRE strains are also among the most common MRD organisms responsible for healthcare-associated infections [30]. The prevalence of VRE in the present study was 4.8%. Prevalences of colonization with VRE in NICUs of 3.6% in the USA, 8.1% in Asia, 12% in Turkey, and 42.2% in Northern Iran have been reported [31,32,33,34]. The prevalence rate in the present study reflects the results of only one center and patients referred from another center in Turkey. This may be due to the fact that our study population consisted of only referred patients, physical differences between the units, and differences in the application of infection control measures. It is therefore difficult to perform a comparison due to different working environments. There are also studies describing prematurity and low birth weight as risk factors for VRE colonization in newborns, and others reporting contradictory viewpoints [33,35,36]. Long hospital stays and the use of broad-spectrum antibiotics have been identified as independent risk factors in this respect [37,38]. Farhadi et al. [34] considered referral from an external healthcare center to be a risk factor. Descriptive features (such as prematurity, antibiotic type and duration, the length of hospital stay, and mechanical ventilator follow-up) had no effects on colonization or non-colonization with either VRE or CRE in patients referred from another healthcare center in the present study (Table 1 and Table 2). A statistically significant association was found between VRE positivity and increased PCT (Table 2). Clinical/definite sepsis, metabolic diseases, asphyxia, and surgical causes were found to be responsible for this elevation, for which VRE could not be implicated as the exact cause. Only one patient with CRE positivity in perirectal swab died, from intestinal perforation and multiorgan failure, and all VRE-positive patients were discharged with recovery.

The rate of positivity in blood culture in newborns is low because of maternal antibiotic use, blood not being collected before starting antibiotics, small volumes of blood being collected for culture, or the absence of bacteremia [39]. A blood culture positivity rate of 15.7% has been reported in children under one year of age with severe sepsis in South Africa (95% confidence interval [CI]: 15.2–16.2%), with rates ranging from 6.6% (95% CI: 6.3–6.9%) to 8.2% (95% CI: 7.9–8.4%) and in two systematic reviews from Africa and Southeast Asia [35,36,37]. In line with the previous literature, blood culture positivity rates at initial hospitalization were low (5.6%) in this study, and no invasive infection was detected with VRE and CRE positivity detected in perirectal swabs. Although the factors identified as CRE-positive in perirectal swab and the microorganisms that reproduced in blood culture were different, perirectal swab screening and the contact isolation surveillance of patients in the external healthcare center may have contributed to this, although it is difficult to reach a definitive conclusion because of the small number of samples involved.

The benefits of microbiological surveillance with universal perirectal and pharyngeal swabs for predicting and preventing sepsis are controversial [9]. Although there are studies reporting a low prognostic value in predicting late neonatal sepsis, others argue that it is very important in endemic regions [9,19]. Folgori et al. [40] showed an association between Gram-negative bacteria (GNB) colonization and confirmed bloodstream infection in newborns with GNB, but were unable to recommend routine perirectal screening because of limited evidence in their NICU. The lack of evidence for the value of perirectal screening for GNB colonization in newborns admitted to NICUs was also confirmed in a meta-analysis in 2017 [41]. Capasso et al. [42] showed that blood culture was identical to the microorganism obtained from a swab in 46 (57%) out of 80 infants with very low birth weights diagnosed with bacterial sepsis. Morom et al. [43] reported that an index case from another healthcare center was capable of causing a silent VRE outbreak in the NICU. In addition to infection prevention and control procedures, those authors recommended the screening of all neonates, both those referred from an external health center and those receiving ongoing treatment in the NICU [43]. The gastrointestinal tract of hospitalized infants has been shown to act as a reservoir for bloodstream infections caused by *Candida spp.*, *Klebsiella spp.*, *E. coli*, and *Pseudomonas spp.* [44]. Das et al. [45] reported a relationship between intestinal colonization and sepsis along with a higher incidence of clinical sepsis in newborns with intestinal GNB colonization compared to those without such colonization. For all these reasons, CRE colonization is an important risk factor for the spread of these bacteria in healthcare settings. The risk of spread in healthcare institutions is high if the asymptomatic carriage of VRE is not detected [46]. The gradual increase in the prevalence of CRE compared to VRE remains an extremely challenging problem in the neonatal population, with higher mortality, morbidity, longer hospital stays, and limited treatment options [47]. Worryingly in the present study, one case of CRE positivity was observed in approximately 4.6 patients and one of VRE positivity in approximately 26 cases. Active perirectal surveillance cultures should be performed and effective control measures adopted in order to detect asymptomatic colonization, since gastrointestinal CRE colonization is also an important risk factor for the spread of these bacteria. This finding also shows that patients referred from external healthcare centers represent an important risk factor by themselves. The systematic approach involving “perirectal swab screening and a contact isolation policy” in our NICU ensured that colonized patients were identified promptly and that effective infection control measures were adopted, with no epidemic occurring in the unit. We also believe that this policy can reduce mortality by preventing invasive infection and cross-dissemination due to microorganisms with MDR. Additionally, the absence of an epidemic in the unit and the fact that positivity was observed only in swab samples probably reflect the compliance of healthcare workers with infection control measures.

Traditional infection control measures in our unit include hand washing, the use of aseptic techniques in interventions, the use of noninvasive mechanical ventilation whenever possible, the use of rational antibiotics, the appropriate design of the intensive care unit, isolation precautions, and especially the use of breast milk. The systematic approach involving “perirectal swab screening and a contact isolation policy for external healthcare center patients”, rather than for all newborns, the use of disposable gowns and gloves in contact with colonized newborns, regular reviews of invasive devices, sufficient bed spacing, appropriate environmental cleaning, and the separation of wastes in the NICU, also yielded significant human, financial and laboratory savings.

The limitations of the present study included the retrospective file scanning and data recording involved and the inability to use molecular analysis methods. Another limitation was that the research was not a randomized controlled study. Had such a randomized controlled trial been performed, we might have more strongly recommended the careful surveillance of CRE and VRE. Larger sample sizes and possibly a tighter surveillance protocol would have yielded stronger results. The particular strength of the research is that it this the first neonatal study to investigate CRE/VRE colonization only in patients hospitalized in an external healthcare center for longer than 48 h.

## 5. Conclusions

This first pilot scheme recommends perirectal screening for MDR organisms, VRE, and CRE, especially in the high-risk group of newborns referred from hospitals where colonizing microorganisms and antibiotic use policies are unclear. The higher prevalence of newborns colonized with CRE is remarkable and must be considered a problem requiring urgent attention both now and in the future. We believe that the absence of an in-unit epidemic with the adoption of this screening method will prepare the ground for multicenter prospective studies with larger sample sizes.

## Figures and Tables

**Figure 1 children-10-00187-f001:**
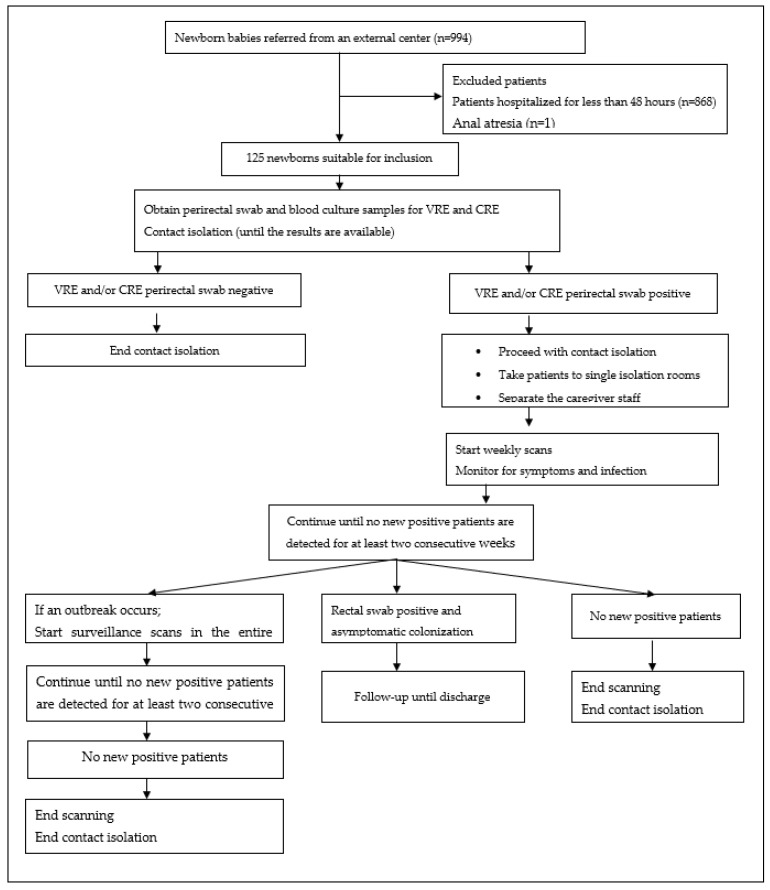
Flow chart of patient selection and methods employed.

**Table 1 children-10-00187-t001:** Relationships between CRE and VRE perirectal swab results and demographic data.

		CRE (Carbapenem-Resistant Enterobacterales)				VRE (Vancomycin-Resistant Enterococci)			
		Positive n (%)	Negative n (%)	Total n (%)	*p* < 0.05	Positive n (%)	Negative n (%)	Total n (%)	*p* < 0.05
Sex				125 (100)	0.436			125 (100)	0.240
	Female	12 (35.3)	41 (45.1)	53 (42.4)		1 (16.7)	52 (43.7)	53 (42.4)	
	Male	22 (64.7)	50 (54.9)	72 (57.6)		5 (83.3)	67 (56.3)	72 (57.6)	
Gestational age					0.932				0.408
	<32 weeks	5 (14.7)	19 (20.9)	24 (19.2)		1 (16.7)	23 (19.3)	24 (19.2)	
	32–36 weeks	15 (44.1)	17 (18.7)	32 (25.6)		2 (33.3)	30 (25.2)	32 (25.6)	
	37 weeks and over	14 (41.2)	55 (60.4)	69 (55.2)		3 (50)	66 (55.5)	69 (55.2)	
* Birth weight (g) (min-max)		2732.5 (2200–3100)	2900 (2240–3400)	-	0.313	2900 (2200–3270)	2780 (2200–3300)	-	0.849
* Hospitalization weight (g)		2745 (2220–3110)	2910 (2100–3370)	-	0.504	2730 (2010–3110)	2850 (2200–3300)	-	0.742
* Hospitalization age—days (min-max)		14 (7–21)	9 (4–18)	-	0.110	7.5 (5–14)	11 (4–20)	-	0.431
Name of Healthcare Center				125 (100)	0.777			125 (100)	1.000
	City of Mersin	28 (82.4)	71 (78)	99 (79.2)		5 (83.3)	94 (79)	99 (79.2)	
	Outside the city	6 (17.6)	20 (22)	26 (20.8)		1 (16.7)	25 (21)	26 (20.8)	
Mode of delivery				125 (100)	0.552			125 (100)	0.176
	Normal delivery	9 (26.5)	31 (34.1)	40 (32.0)		0 (0)	40 (33.6)	40 (32.0)	
	Cesarean	25 (73.5)	60 (65.9)	85 (68.0)		6 (100)	79 (66.4)	85 (68)	
External center antibiotherapy				125 (100)	0.165			125 (100)	0.867
	No antibiotic treatment	9 (26.5)	18 (19.8)	27 (21.6)		1 (16.7)	26 (21.8)	27 (21.6)	
	Empirical (ampicillin- gentamicin)	13 (38.2)	49 (53.8)	62 (49.6)		3 (50)	59 (49.6)	62 (49.6)	
	^#^ Broad-spectrum antibiotic	12 (35.3)	24 (26.4)	36 (28.8)		2 (33.3)	34 (28.6)	36 (28.8)	
Externalcenter antibiotherapy duration				125 (100)	0.764			125 (100)	0.634
	0 days	9 (26.5)	17 (18.7)	26 (20.8)		9 (25)	18 (20.2)	27 (21.6)	
	1–10 days	19 (55.9)	68 (74.7)	87 (69.6)		21 (58.3)	65 (73.0)	86 (68.8)	
	>10 days	6 (17.6)	6 (6.6)	12 (9.6)		6 (16.7)	6 (6.7)	12 (9.6)	

* Median (minimum and maximum) values are given instead of “n” and “%”, ^#^ Carbapenem + Vancomycin ± Amikacin ± Antifungal therapy, Abbreviations: CRE, carbapenem-resistant *Entrobacterales*; VRE, vancomycin-resistant *Enterococci*.

**Table 2 children-10-00187-t002:** Relationships between CRE and VRE results in perirectal swabs and patient clinical characteristics after admission to the unit.

		CRE (Carbapenem-Resistant Enterobacterales)				VRE (Vancomycin-Resistant Enterococci)			
		Positive n (%)	Negative n (%)	Total n (%)	*p* < 0.05	Positive n (%)	Negative n (%)	Total n (%)	*p* < 0.05
Duration of hospitalization after referral				125 (100)	0.089			125 (100)	0.039
	1–10 days	7 (20.6)	28 (30.8)	35 (28.0)		3 (50)	32 (26.9)	35 (28.0)	
	11–20 days	6 (17.6)	24 (26.4)	30 (24.0)		3 (50)	27 (22.7)	30 (24.0)	
	>20 days	21 (61.8)	39 (42.9)	60 (48.0)		0 (0)	60 (50.4)	60 (48.0)	
Mechanical ventilator support				125 (100)	1.000		125 (100)		0.401
	Yes	15 (44.1)	39 (42.9)	54 (43.2)		4 (66.7)	50 (42)	54 (43.2)	
	No	19 (55.9)	52 (57.1)	71 (41.6)		2 (33.3)	69 (58)	71 (56.8)	
Sepsis				125 (100)	0.818			125 (100)	0.895
	None	16 (47.1)	41 (45.1)	57 (45.6)		2 (33.3)	55 (46.2)	57 (45.6)	
	Clinical sepsis	11 (32.4)	32 (35.2)	43 (34.4)		3 (50)	40 (33.6)	43 (34.4)	
	Proven sepsis	3 (8.8)	4 (4.4)	7 (5.6)		1 (16.7)	6 (5)	7 (5.6)	
	Suspicious sepsis	4 (11.8)	14 (15.4)	18 (14.4)		0 (0)	18 (15.1)	18 (14.4)	
* Laboratory									
	White blood cell (mm^3^)	11,510 (8800–16,290)	10,770 (8340–14,200)	-	0.451	9220 (8870–147,20)	11,430 (8350–14,550)	-	0.817
	C-reactive protein (mg/L)	1.7 (0.5–12.5)	2.7 (0.7–15.5)	-	0.564	7 (1.4–12)	2.5 (0.6–15.5)	-	0.564
	Procalcitonin (ng/dL)	2.4 (0.8–3.4)	2 (0.7–3.5)	-	0.814	3.5 (2.6–3.8)	2 (0.7–3.2)	-	0.039
	Band-neutrophil ratio	0.2 (0.1–0.2)	0.1 (0.1–0.2)	-	0.619	0.2 (0.2–0.3)	0.1 (0.1–0.2)	-	0.087
Blood culture factor				125 (100)	0.323			125 (100)	0.038
	No growth	31 (91.2)	87 (95.6)	118 (94.4)		5 (83.3)	113 (95)	118 (94.4)	
	MRCNS	2 (5.9)	3 (3.3)	5 (4.0)		0 (0)	5 (4.2)	5 (4.0)	
	*K. pneumonia*	1 (2.9)	1 (1.1)	2 (1.6)		1 (16.7)	1 (0.8)	2 (1.6)	
Mortality				125 (100)	0.206			125 (100)	0.461
	No	33 (97.1)	82 (90.1)	115 (92.0)		6 (100)	109 (91.6)	115 (92.0)	
	Yes	1 (2.9)	9 (9.9)	10 (8.0)		0 (0.0)	10 (8.1)	10 (8.0)	

* Median (minimum and maximum) values are given instead of “n” and “%”. Abbreviations: CRE, carbapenem-resistant *Enterobacteriales*; VRE, vancomycin-resistant *Enterococci*; MRCNS, methicillin resistant coagulase-negative *Staphylococci*; *K. Pneumonia*, *Klebsiella pneumonia*.

**Table 3 children-10-00187-t003:** Hospitalization causes and perirectal swab results.

Definite Diagnosis	Rectal Swab		
	Positive n (%)	Negative n (%)	*p* < 0.05
Respiratory distress	5 (13.9)	26 (29.2)	0.391
Metabolic causes	7 (19.4)	16 (18.0)	
Surgical causes	9 (25)	14 (15.7)	
Asphyxia/Neonatal convulsion	7 (19.4)	14 (15.7)	
Sepsis	6 (16.7)	10 (11.2)	
Hematological/oncological causes	2 (5.6)	9 (10.1)	

## Data Availability

The data presented in this study are available on request from the corresponding author.

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
