# Peer review of "Should Perirectal Swab Culture Be Performed in Cases Admitted to the Neonatal Intensive Care Unit? Lessons Learned from the Neonatal Intensive Care Unit"

_children, 2023, doi:10.3390/children10020187_

Round 1
Reviewer 1 Report
I am a neonatologist working in a tertiary unit and I agree with the perirectal swab culture. We also take culture swabs from nose, ear and skin. It is a way to keep safe the NICU environment .
It is well known that multi drug resistant bacteria are a nightmare for neonatologists. Also this kind of bacteria can be spread very easily inside NICU. So every method of prevention is good.
Author Response
DeÄŸerli vaktinizi bu makaleyi incelemek üzere ayırdığınız için çok teÅŸekkür ederiz.
El yazmasının tüm metni anadili ile revize edildi.

Reviewer 2 Report
This manuscript reports the Authors’ experience with perirectal swabs to test outpatients entering the NICU for CRE and VRE, which represent a growing threat in some areas of the world. This is a hot topic, also considering there is a lack of consensus on the regular use of surveillance swabs to detect colonized patients.
Please, find my comments and suggestions below.
Introduction: Well-written, references adequate.
Materials and Methods:
1. The first part of this section needs to be re-organized and developed, as follows:
- First paragraph:” Study Design, Setting, and Population”
- Second paragraph: “Case definition and Inclusion criteria”
- Third paragraph: “Data collection”
- Fourth paragraph: “Outcomes”
- Then the already existing paragraphs with “Microbiological evaluation” and “statistical analysis”
2. Could you please describe how your NICU is organized? Are the 27 cots all in an open-space area, or are there separate rooms? Are there any isolation, single-cot, rooms?
3. What is the Doctor/patient ratio?
4. Is it the normal procedure at your facility to not send samples for general culture for all MDR organisms, including ESBL? Do you always test only for VRE and CRE?
5. In case a baby was colonized, were contact precautions also used by the parents when visiting, or only by the staff?
6. You use “perianal” and “perirectal” indifferently throughout the manuscript. Please specify if you employed perianal (cutaneous) or rectal (mucous) swabs for culture, and correct accordingly. Moreover, do you confirm that you did not use fecal samples for culture?
7. Did you also test the staff? If so, any positive results and how did you manage the situation? Please describe the conventional IPC practices at your facility (line 298 in Discussion?)
8. Did not you also collect air and water samples? If not, why? What were the results from environmental samples?
Results:
9. Line 165: “Perirectal swab positivity was 27.2% CRE and 4.8% VRE.” Please provide also numbers, along with percentages.
10. Please also provide the total number of positive and negative patients for CRE and VRE, respectively, in the tables. Results are otherwise difficult to interpret.
11. Line 166: “Analysis revealed no significant relationship between RESPECTIVELY CRE and VRE results in perirectal swab and demographic data”: please rephrase accordingly for clarity. You mean that the groups were homogenous, right?
12. Please, add a paragraph where you explicitly report results for the outcomes you stated in the results section.
Discussion:
In general, I advise you to try and correlate better your results with the available literature for newborns in NICUs. The available data from the literature are otherwise well described.
13. From Line 221: This study suggests that although the infrastructure and current problems of the units to which the patients are referred are not known, the high prevalence of CRE may be related to the high rate of long-term and unindicated use of carbapenems and that infection prevention and control measures need to be reviewed”. This statement is probably true, but you can’t say that you get that from your study's results, which show no statistically significant differences between CRE positive and negative patients, regarding previous use of broad-spectrum antibiotics/carbapenems. I advise you to keep this concept but rephrase it.
14. Line 226_ “Length of hospital stay, orogastric nutrition, exposure to antipseudomonal antibiotics, previous surgical operation, and mechanical ventilation have been identified as separate risk factors for CRE colonization”. May you please comment on these findings from the literature, in comparison with the results of your study, which show no statistically significant differences between positive and negative patients for a longer length of stay for positive patients and for exposure to antipseudomonal antibiotics and mechanical ventilation?
15. Line 248 “The rate of positivity in blood culture in newborns is low because of maternal antibiotic use, blood not being collected before starting antibiotics, taking small volumes of blood for culture, or lack of bacteremia”. Please rephrase, ie: “The rate of positivity […] may be due to…”.
16. Please state Limitations in a dedicated paragraph.
17. Conclusions: In line 313, you write “incidence”, but you deal with prevalence throughout the study. Please correct accordingly.
Author Response
Dear Reviewer
Thank you very much for giving up your precious time to review this article.
The entire text of the manuscript has been revised with native speaker.
In line with your suggestions;
Materials and Methods: The first part of this section has been arranged as follows:
- First paragraph:” Study Design, Setting, and Population”
- Second paragraph: “Case definition and Inclusion criteria”
- Third paragraph: “Data collection”
- Fourth paragraph: “Outcomes”
- Could you please describe how your NICU is organized? Are the 27 cots all in an open-space area, or are there separate rooms? Are there any isolation, single-cot, rooms?
There are 25 beds in in an open area in our unit, plus two single isolation rooms. Two neonatology specialists (a professor of medicine and an assistant professor of edicine), a neonatology intern and three pediatric interns work in our NICU.
- What is the Doctor/patient ratio?
The doctor/patient ratio is 1/7
- 4.Is it the normal procedure at your facility to not send samples for general culture for all MDR organisms, including ESBL? Do you always test only for VRE and CRE?
We routinely screen only for CRE and VRE in our unit. However, if an extraordinary multidrug resustant micro-organism is determined in blood culture, than all hopsitalized patients can be screened as recommended by the infection committee.
- In case a baby was colonized, were contact precautions also used by the parents when visiting, or only by the staff?
In the presence of a colonized infant, the parents also implemented contact precautions.
- 6.You use “perianal” and “perirectal” indifferently throughout the manuscript. Please specify if you employed perianal (cutaneous) or rectal (mucous) swabs for culture, and correct accordingly. Moreover, do you confirm that you did not use fecal samples for culture?
The word perianal in the manuscript has been corrected to 'perirectal' . Stool samples were not used for further culture.
- Did you also test the staff? If so, any positive results and how did you manage the situation? Please describe the conventional IPC practices at your facility (line 298 in Discussion?)
Staphylococcal screening was not performed. For this reason, we cannot make any comments about the results.
Traditional IPC applications in our facility are as follows;
- Did not you also collect air and water samples? If not, why? What were the results from environmental samples?
We did not collect air and water samples. No outbreaks related to CRE or VRE were detected in the unit during the study period. As part of the routine protocol of our unit, perirectal swab scanning and contact isolation are still mintained for patients who are referred from an external center.
Since VRE is capable of surviving on inanimate surfaces for very lengthy periods, it is recommended that media cultures be taken on every ward where VRE growth is detected in perirectal culture.
Media cultures are taken from the room of the patient or patients with VRE growth and from other surfaces in that ward.
Cultures should especially be taken from surfaces that are in frequent contact with hands (door handles, nightstands, dining tables, doctor and nurse observation tables, IV poles, faucets, monitors, ventilators, etc.) Which surfaces will be cultured is determined by the Infection Control Team, and culture is performed by the same team.
The same protocol is followed for CRE when an outbreak is suspected. It is not recommended that samples be taken from the environment and inanimate surfaces, except for studies to be performed for source detection in case of an epidemic with CRE.
Results:
- 9.Line 165: “Perirectal swab positivity was 27.2% CRE and 4.8% VRE.” Please provide also numbers, along with percentages.
The numbers have also been provided along with the percentages (27.2% (n=34), 4.8 VRE (n=6)
- Please also provide the total number of positive and negative patients for CRE and VRE, respectively, in the tables. Results are otherwise difficult to interpret.
The total number of patients for CRE and VRE has been added to the tables (Table 1 and Table 2).
- Line 166: “Analysis revealed no significant relationship between RESPECTIVELY CRE and VRE results in perirectal swab and demographic data”: please rephrase accordingly for clarity. You mean that the groups were homogenous, right?
Perirectal swab results were divided into groups - CRE positive and negative, and VRE positive and negative. The relationship between these groups and demographic data was investigated. No statistically significant difference was found. p values are given in Table 1. The groups were not homogeneous.
- Please, add a paragraph where you explicitly report results for the outcomes you stated in the results section.
A summary paragraph has been added in which we report the results.
Outcome; Perirectal swab positivity rates were (28.8%, n=36) in the study population. Perirectal CRE positivity (27.2%) and VRE positivity (4.8%) were detected. No proven sepsis due to these microorganisms and no epidemic in the unit were detected.
Discussion:
In general, I advise you to try and correlate better your results with the available literature for newborns in NICUs. The available data from the literature are otherwise well described.
- From Line 221: This study suggests that although the infrastructure and current problems of the units to which the patients are referred are not known, the high prevalence of CRE may be related to the high rate of long-term and unindicated use of carbapenems and that infection prevention and control measures need to be reviewed”. This statement is probably true, but you can’t say that you get that from your study's results, which show no statistically significant differences between CRE positive and negative patients, regarding previous use of broad-spectrum antibiotics/carbapenems. I advise you to keep this concept but rephrase it.
- Line 226_ “Length of hospital stay, orogastric nutrition, exposure to antipseudomonal antibiotics, previous surgical operation, and mechanical ventilation have been identified as separate risk factors for CRE colonization”. May you please comment on these findings from the literature, in comparison with the results of your study, which show no statistically significant differences between positive and negative patients for a longer length of stay for positive patients and for exposure to antipseudomonal antibiotics and mechanical ventilation?
n this study, descriptive features such as exposure and duration of broad-spectrum antibiotics, length of hospital stay, mechanical ventilator follow-up were observed to have no effect on colonization or non-colonization.
- 15.Line 248 “The rate of positivity in blood culture in newborns is low because of maternal antibiotic use, blood not being collected before starting antibiotics, taking small volumes of blood for culture, or lack of bacteremia”. Please rephrase, ie: “The rate of positivity […] may be due to…”.
- Please state Limitations in a dedicated paragraph.
Limitations have been given in a special paragraph as suggested.
- Conclusions: In line 313, you write “incidence”, but you deal with prevalence throughout the study. Please correct accordingly.
The word incidence has been corrected as prevalence.

Reviewer 3 Report
In this paper, the Authors aimed to assess the carbapenem-resistant Enterobacterales (CRE) and vancomycin-resistant Enterococci (VRE) colonization positivity rate of newborns previously hospitalized in external hospital centers for more than 48 hours, in order to prevent MDR-related sepsis and spread within the neonatal intensive care units (NICU). The aim gets me a bit confused, and I demand the Authors be clear.
The English form should be ameliorated, as well as the soundness of the paper.
The statistical analyses seem to be well-conducted with the usage of proper tests.
I have several comments and suggestions for the Authors that need to be addressed.
Abstract
Lines 23-28: Try to rephrase, because it is difficult to understand what the Authors meant. Moreover, I would say “Additional purpose” instead of “Another aim”.
Line 29: what do the Authors mean by “this”? It is necessary to specify the subject of the sentence.
Lines 32-33: I would say “one out of every” instead of “one in every”.
Introduction
In general, the introduction seems to be exhaustive, but it needs to be more punctual and carefully revised. Nonetheless:
Lines 45-46: “Outbreaks of sepsis due to MDR agents seem to be potentially responsible for approximately 30% of neonatal mortality”. The reference is from 2016, just like the previous ones but, in this case, this suggests that the information might be outdated. I suggest more recent data and adding newly published manuscripts as references.
Line 47: I do not find the word “suspicion” adequate to the context. Please, reword it.
Lines 48-52: I suggest this rephrased version: “To prevent infection or colonization by MDR organisms, it is necessary to combine hand hygiene monitoring systems, MDR agent surveillance, contact isolation measures (e.g., single room isolation or group cohorts of cases exposed to bacterial colonization), environmental cleanliness, prudent antimicrobial agent use, and antibiotic management strategies.” Moreover, I would alternate the use of “agents” and “organisms” or other synonyms.
Lines 52-54: This sentence needs a reference.
Lines 54-58: This paragraph is hard to comprehend. Moreover, how is the reference correlated to the part in which the Authors talk about Turkey?
Line 59: “Pseudomonas aeruginosa”.
Lines 61-64: Please, rephrase this sentence.
Lines 64-65: I would say that “rectal swab collection is performed as part of surveillance”.
Lines 70-71: “Therefore, […]”
Lines 71-72: If the Authors want to speak about the costs of processing swabs
Lines 72-77: I would start a new line. Moreover, it is not clear what are the aims of the study: the purposes of the manuscript must be acknowledged to the Readers in a precise and timely form. This is a very crucial part. Lastly, are the Authors saying that they normally test just for CRE and VRE colonization? Isn’t it too restrictive? I think it is important to test for other important colonization bacteria and fungi and I believe that it routinely happens.
Materials and Methods
Lines 80-82: This part is a repetition of what has been correctly written in Lines 323-325. I suggest the removal of this sentence.
Lines 93-96: These are not just demographic data (e.g. birth weight, antibiotic treatment), they should be defined as “demographic and anamnestic/clinical data”. Moreover, be careful not to misuse the term gender: use the term sex, instead. Reword all along the manuscript.
Lines 86-128: Since it is rich with details and at the very least needs to be schematic, I recommend the authors to include all the inclusion/exclusion criteria, as well as the procedures used when a certain event occurred and the definition of cases, in a flowchart.
Statistical analysis
Lines 152-153: “Whether the continuous variables were compatible with normal distribution was also checked”. How was it checked? Which test did the Authors perform?
Results
Line 163: why were the perirectal swabs exactly double the enrolled participants? Please, explain this in the manuscript.
Table 1: I advise the Authors to use the extended terms for CRE and VRE. Moreover, center the p-values (same advice for Tables 2 and 3).
Discussion
The Discussion seems to be rich in content, but it needs moderate English corrections. By the way, instead of commenting on the outcomes ex abrupto, I would write something to present the initial discussion point.
Lines 231-234: it seems that the sample is not representative of the prevalence of VRE in Turkey. It should be underlined as a limit, as well as the others that the Authors have already written about.
Moreover, I think the Authors should comment on the non-significativity of the results, especially when discussing mortality. VRE and particularly CRE colonization might be a serious threat, but the results apparently do not support the necessity for careful surveillance. The Authors should discuss this part also in the limitations paragraph.
Author Response
Dear Reviewer
Thank you very much for giving up your precious time to review this article.
The entire text of the manuscript has been revised with native speaker.
Abstract
Lines 23-28: Try to rephrase, because it is difficult to understand what the Authors meant. Moreover, I would say “Additional purpose” instead of “Another aim”.
This has been amended according to your suggestions.
Additional purpose" has replaced "Another aim’’
Line 29: what do the Authors mean by “this”? It is necessary to specify the subject of the sentence.
Lines 32-33: I would say “one out of every” instead of “one in every”.
Answer; Amended as suggested in line with your suggestions, the sentence has been changed.
‘‘…..that ‘one out of every’ 4.4 infants included in the study had perirectal swab positivity.’’
Introduction
In general, the introduction seems to be exhaustive, but it needs to be more punctual and carefully revised. Nonetheless:
Answer; The introduction has been revised as suggested.
Lines 45-46: “Outbreaks of sepsis due to MDR agents seem to be potentially responsible for approximately 30% of neonatal mortality”. The reference is from 2016, just like the previous ones but, in this case, this suggests that the information might be outdated. I suggest more recent data and adding newly published manuscripts as references.
Based on population-level studies over the past two decades, the global estimated incidence of mortality from neonatal sepsis ranges from 11% to 19%. Due to their increasing incidence, MDR microorganisms are estimated to be responsible for a significant proportion of deaths from neonatal sepsis.( Dong, Y.; Basmaci, R.; Titomanlio, L.; Sun, B., Mercier, J. C. Neonatal sepsis: within and beyond China. Chin Med J (Engl) 2020, 133, 2219-2228. doi:10.1097/cm9.0000000000000935)
Answer;
Line 47: I do not find the word “suspicion” adequate to the context. Please, reword it.
Lines 48-52: I suggest this rephrased version: “To prevent infection or colonization by MDR organisms, it is necessary to combine hand hygiene monitoring systems, MDR agent surveillance, contact isolation measures (e.g., single room isolation or group cohorts of cases exposed to bacterial colonization), environmental cleanliness, prudent antimicrobial agent use, and antibiotic management strategies.” Moreover, I would alternate the use of “agents” and “organisms” or other synonyms.
Answer;
The following paragraph has been added to the manuscript” “To prevent infection or colonization by MDR organisms, it is necessary to combine hand hygiene monitoring systems, MDR agent surveillance, contact isolation measures (e.g., single room isolation or group cohorts of cases exposed to bacterial colonization), environmental cleanliness, prudent antimicrobial agent use, and antibiotic management strategies’.
Lines 52-54: This sentence needs a reference.
Answer: The reference has been added
Serial screening of selected body sites with swab cultures is a common practice in NICUs to monitor horizontal pathogen spread and prevent the occurrence of septic outbreaks ‘’Saporito L, Graziano G, Mescolo F, Amodio E, Insinga V, Rinaudo G, Aleo A, Bonura C, Vitaliti M, Corsello G, Vitale F, Maida CM, Giuffrè M. Efficacy of a coordinated strategy for containment of multidrug-resistant Gram-negative bacteria carriage in a Neonatal Intensive Care Unit in the context of an active surveillance program. Antimicrob Resist Infect Control. 2021 Feb 4;10(1):30. doi: 10.1186/s13756-021-00902-1. PMID: 33541419; PMCID: PMC7863509’’
Lines 54-58: This paragraph is hard to comprehend. Moreover, how is the reference correlated to the part in which the Authors talk about Turkey?
Answer:
Over time, changes have been observed in multi-drug resistant infectious agents worldwide and in Turkey, and new antibiotics effective against gram positive have entered into use.Due to these new changes, there have been doubts about whether surveillance prevents increases in VRE and causes changes in mortality and morbidity rates .
An article from Turkey has been added as a reference.
(Ulu-Kilic A, Özhan E, Altun D, Perçin D, GüneÅŸ T, Alp E. Is it worth screening for vancomycin-resistant Enterococcus faecium colonization?: Financial burden of screening in a developing country. Am J Infect Control. 2016 Apr 1;44(4):e45-9. doi: 10.1016/j.ajic.2015.11:008. Epub 2016 Jan 5. PMID: 26775930.)
Line 59: “Pseudomonas aeruginosa”.
Answer : This has been corrected to “Pseudomonas aeruginosa”
Lines 61-64: Please, rephrase this sentence.
Answer : The sentence has been reworded
‘’ Despite this decrease in Gram-positive agents, VRE still causes widespread colonization in the early period and invasive and difficult-to-treat infections in critically ill patients by causing in-hospital transmission from patients and the environment’’
Lines 64-65: I would say that “rectal swab collection is performed as part of surveillance”.
Answer: This sentence has been reworded.
‘Rectal swab collection may therefore be recommended as part of surveillance in critically hospitalized patients’
Lines 70-71: “Therefore, […]”
Lines 71-72: If the Authors want to speak about the costs of processing swabs
However, whether the screening strategy is a cost-effective measure remains a matter of debate. Institutions in developing countries should identify their own high-risk patients, and screening priorities should be based on infection prevalence and hospital finances
Lines 72-77: I would start a new line. Moreover, it is not clear what are the aims of the study: the purposes of the manuscript must be acknowledged to the Readers in a precise and timely form. This is a very crucial part. Lastly, are the Authors saying that they normally test just for CRE and VRE colonization? Isn’t it too restrictive? I think it is important to test for other important colonization bacteria and fungi and I believe that it routinely happens.
Answer : The purpose has been reworded in a more understandable way.
In this study, VRE and CRE colonization were routinely investigated in patients admitted from an external center and hospitalized in our neonatal intensive care unit. The purpose of this study was to prevent the spread of VRE and CRE by applying contact isolation to screened patients. This is the first pilot study to evaluate the spread and epidemic of sepsis within the NICU using this method
Materials and Methods
Lines 80-82: This part is a repetition of what has been correctly written in Lines 323-325. I suggest the removal of this sentence.
Answer : This sentence has been removed in line with your suggestions
Lines 93-96: These are not just demographic data (e.g. birth weight, antibiotic treatment), they should be defined as “demographic and anamnestic/clinical data”. Moreover, be careful not to misuse the term gender: use the term sex, instead. Reword all along the manuscript.
Answer: Amended to “demographic and anamnestic/clinical data” as suggested. The term sex has been used instead of.
Lines 86-128: Since it is rich with details and at the very least needs to be schematic, I recommend the authors to include all the inclusion/exclusion criteria, as well as the procedures used when a certain event occurred and the definition of cases, in a flowchart.
A flow chart containing the study design was created
Statistical analysis
Lines 152-153: “Whether the continuous variables were compatible with normal distribution was also checked”. How was it checked? Which test did the Authors perform?
Answers: The normality of the distribution of continuous measurements was checked using the Kolmogorov-Smirnov test.
Results
Line 163: why were the perirectal swabs exactly double the enrolled participants? Please, explain this in the manuscript.
Answer: Perirectal swab samples for CRE and VRE were collected from patients who met the inclusion criteria within the first 24 hours of admission to our unit by a trained infection nurse using a sterile cotton swab moistened with 0.9% NaCl. Two hundred fifty swab cultures were thus obtained from 125 patients. An explanation has been added to the microbiological evaluation section.
Table 1: I advise the Authors to use the extended terms for CRE and VRE. Moreover, center the p-values (same advice for Tables 2 and 3).
Answer : Extended terms for CRE and VRE have been added to the tables, and the p-values have been centered. (Tables 2 and 3).
Discussion
The Discussion seems to be rich in content, but it needs moderate English corrections. By the way, instead of commenting on the outcomes ex abrupto, I would write something to present the initial discussion point.
Answer: A paragraph has been added to present the first discussion point.
Perirectal active surveillance of MDR microorganism carriage can be considered an effective tool for the early detection of unusual microbial pathogen patterns.
Emerging evidence indicates an increased prevalence of MDR gram-negative (MDR-GN) colonization and neonatal MDR-GN infection in neonatal settings.
Lines 231-234: it seems that the sample is not representative of the prevalence of VRE in Turkey. It should be underlined as a limit, as well as the others that the Authors have already written about.
Moreover, I think the Authors should comment on the non-significativity of the results, especially when discussing mortality. VRE and particularly CRE colonization might be a serious threat, but the results apparently do not support the necessity for careful surveillance. The Authors should discuss this part also in the limitations paragraph.
Answer; Another limitation was that this research was not a randomized controlled study. Had such a randomized controlled trial been performed, we might have more strongly recommended careful surveillance of CRE and VRE.

Round 2
Reviewer 2 Report
Dear Authors,
Thanks for providing the updated version of the manuscript and for addressing my comments and suggestions.
The paper in its present form is substantially improved. Well done!
I have some last minor comments for you to address:
- Lines 102-104: Ethics information must be included in the Methods section (do not delete it!), you may include it in the 2.1 Paragraph.
- Figure 1: The arrow indicating the excluded patients should be facings outwards (and not inwards).
- Lines 173-178 “CRE and/or VRE colonization in this study was defined as cases with positive perirectal swab samples without signs or symptoms of infection in the newborn. CRE and/or VRE infection was defined as a positive sample from a normally sterile site (blood and cerebrospinal fluid) with signs and symptoms of infection. The definition of the outbreak was based on isolates of the same species from two or more sterile sites with the same antibiogram from different patients over two weeks.”
This part does not refer to the outcomes and should therefore be moved to the 2.1 Paragraph.
- Line 259: Please, correct as follows: “VRE-negative patients at the perirectal swab…”
- It would be very helpful to add also the grand total to the tables, which are much improved and readable after modification.
Author Response
Dear Reviewer
Thank you very much for giving up your precious time to review this article.
Lines 102-104: Ethics information must be included in the Methods section (do not delete it!), you may include it in the 2.1 Paragraph.
Answer: Ethical information has been added to the Methods section.
- Figure 1: The arrow indicating the excluded patients should be facings outwards (and not inwards).
Answer : The arrow pointing out the excluded patients is corrected outward.
- Lines 173-178 “CRE and/or VRE colonization in this study was defined as cases with positive perirectal swab samples without signs or symptoms of infection in the newborn. CRE and/or VRE infection was defined as a positive sample from a normally sterile site (blood and cerebrospinal fluid) with signs and symptoms of infection. The definition of the outbreak was based on isolates of the same species from two or more sterile sites with the same antibiogram from different patients over two weeks.”
This part does not refer to the outcomes and should therefore be moved to the 2.1 Paragraph.
Answer : This section has been moved to Paragraph 2.1.
- Line 259: Please, correct as follows: “VRE-negative patients at the perirectal swab…”
Answer: The sentence as “VRE negative patients on perirectal swab” has been corrected.
- It would be very helpful to add also the grand total to the tables, which are much improved and readable after modification.
Answer: The 'grand total' has been added to the tables
Reviewer 3 Report
The Authors, in my perspective, accomplished good work; I am delighted to notice that they increased the manuscript's quality and that it is now significantly more organized.
At last, I recommend revising the abstract. Something is missing: the Authors began by discussing the aims of this article, but they should provide some context, perhaps with a quick introduction sentence or two.
Author Response
Dear Reviewer
Thank you very much for giving up your precious time to review this article.
At last, I recommend revising the abstract. Something is missing: the Authors began by discussing the aims of this article, but they should provide some context, perhaps with a quick introduction sentence or two.
Answer: A binding sentence has been added to the abstract section before the purpose.
‘’Serial perirectal swabs are used to identify colonization of multidrug-resistant bacteria and prevent spread.’’
The purpose of this study was to determine colonization with carbapenem-resistant Enterobacterales (CRE) and vancomycin-resistant Enterococci (VRE). An additional purpose was to establish whether sepsis and epidemic associated with these factors were present in the neonatal intensive care unit (NICU) to which infants with hospital stays exceeding 48 hours in an external healthcare center NICU were admitted. Perirectal swab samples were collected in the first 24 hours by a trained infection nurse using sterile cotton swabs moistened with 0.9% NaCl from patients admitted to our unit after hospitalization exceeding 48 hours in an external center. The primary outcome was positivity in perirectal swab cultures, and the secondary outcomes were whether this caused invasive infection and significant NICU outbreaks. A total of 125 newborns meeting the study criteria referred from external healthcare centers between January 2018 and January 2022 were enrolled. Analysis revealed that CRE constituted 27.2% of perirectal swab positivity and VRE 4.8%, and that one in every 4.4 infants included in the study exhibited perirectal swab positivity. Detection of colonization by these microorganisms and including them within the scope of surveillance is an important factor in the prevention of NICU epidemics.’